# Smart Drug-Delivery System of Upconversion Nanoparticles Coated with Mesoporous Silica for Controlled Release

**DOI:** 10.3390/pharmaceutics15010089

**Published:** 2022-12-27

**Authors:** Yanan Huang, Ziqing Du, Guochen Bao, Guocheng Fang, Matthew Cappadona, Lana McClements, Bernard E. Tuch, Hongxu Lu, Xiaoxue Xu

**Affiliations:** 1School of Mathematical and Physical Sciences, Faculty of Science, University of Technology Sydney, Sydney, NSW 2007, Australia; 2Department of Chemistry, Shanghai Key Lab of Molecular Catalysis and Innovative Materials, iChEM, Fudan University, Shanghai 200437, China; 3School of Life Sciences & Institute for Biomedical Materials and Devices, Faculty of Science, University of Technology Sydney, Sydney, NSW 2007, Australia; 4Australia Foundation for Diabetes Research, Sydney, NSW 2007, Australia; 5Department of Diabetes, Central Clinical School, Faculty of Medicine, Nursing & Health Sciences, Monash University, Melbourne, VIC 3800, Australia; 6State Key Laboratory of High Performance Ceramics and Superfine Microstructure, Shanghai Institute of Ceramics, Chinese Academy of Sciences, 1295 Dingxi Road, Shanghai 200050, China; 7School of Biomedical Engineering, Faculty of Engineering and Information Technology, University of Technology Sydney, Sydney, NSW 2007, Australia

**Keywords:** upconversion nanoparticles, mesoporous silica, lysozyme, drug delivery, near-infrared light

## Abstract

Drug-delivery vehicles have garnered immense interest in recent years due to unparalleled progress made in material science and nanomedicine. However, the development of stimuli-responsive devices with controllable drug-release systems (DRSs) is still in its nascent stage. In this paper, we designed a two-way controlled drug-release system that can be promoted and prolonged, using the external stimulation of near-infrared light (NIR) and protein coating. A hierarchical nanostructure was fabricated using upconversion nanoparticles (UCNPs)—mesoporous silica as the core-shell structure with protein lysozyme coating. The mesoporous silica shell provides abundant pores for the loading of drug molecules and a specific type of photosensitive molecules. The morphology and the physical properties of the nanostructures were thoroughly characterized. The results exhibited the uniform core-shell nanostructures of ~four UCNPs encapsulated in one mesoporous silica nanoparticle. The core-shell nanoparticles were in the spherical shape with an average size of 200 nm, average surface area of 446.54 m^2^/g, and pore size of 4.6 nm. Using doxorubicin (DOX), a chemotherapy agent as the drug model, we demonstrated that a novel DRS with capacity of smart modulation to promote or inhibit the drug release under NIR light and protein coating, respectively. Further, we demonstrated the therapeutic effect of the designed DRSs using breast cancer cells. The reported novel controlled DRS with dual functionality could have a promising potential for chemotherapy treatment of solid cancers.

## 1. Introduction

Spurred by the recent advances in material and biological sciences, “smart” stimuli-responsive drug-release systems (DRSs) are rapidly emerging platforms of choice for controlled chemotherapy drug delivery. The stimuli-responsive DRSs commonly consist of two or more functional materials for carrying drugs that work synergistically, responding and adapting to stimuli, and releasing drugs in a spatial, temporal, and dosage-controlled manner. The execution of such stimuli-responsive DRSs hinges upon two critical components: (i) a biocompatible nanocarrier with high drug loading amount and (ii) the incorporated sensitive and efficient stimuli-responsive valve.

While a variety of biocompatible and/or biodegradable drug nanocarriers, such as polymers and liposomes [1,2,3,4,5], have been explored for DRS design, mesoporous silica-containing nanoparticles (MSNs) have shown the most promise. This is because of their extreme high porosity that enables high dose drug loading and versatile surface that allows for various surface functionalization to incorporate stimuli-responsive molecules [6,7,8,9]. In terms of the stimuli, light is particularly attractive compared to pH- [10], temperature- [11], or enzyme-based designs [12,13,14] because, as an external stimulus, light stimulation with feasibly tunable light wavelength and input power could offer both spatially and temporally controllable drug release. Particularly, near-infrared (NIR, 750–1000 nm) light is the most promising as external stimulation because it exhibits low photocytotoxicity and improved tissue penetration depth over the ultraviolet (UV) or visible light [15]. The NIR light-responsive nanomedicine systems could achieve spatiotemporal, pulsatile, and on-demand drug release at targeted locations [16,17,18], enabling their appeal as a non-invasive technology for various clinical applications [15,19].

Current NIR light-induced photochemistry- and/or phototherapy-based technologies for DRSs development largely rely upon upconversion nanoparticles (UCNPs) [20,21], which serve as nanotransducers, absorbing single-band NIR light and subsequently emitting high-energy UV-to-visible light. UCNPs possess unique photonic properties, including distinct NIR light excitation, sharp emission peaks, high photostability, low cytotoxicity, low photodamage, and zero-autofluorescence, deeming them the unique candidate for the NIR light-responsive DRSs [22,23,24,25]. Qu et al. anchored photochromic spiropyran on the mesoporous silica-coated UCNPs as an NIR-controlled DRS and achieved greatly enhanced therapeutic efficiency [26]. Yang et al. demonstrated a good release profile of an anti-cancer drug, doxorubicin (DOX), from mesoporous silica-coated Ln-UCNPs; upon irradiation, the DOX can be released from the light-opened capsule [27]. Although the UCNPs-based DRSs have been widely explored [24,25], the dual-function DRSs that can both trigger and slow down drug release within one single nano-system are still rare. Paula et al. reported protein binding on the surface of MSNs, which blocks mesopores and then delays the drug release at low drug loadings (3 wt%) in vitro [28]. In contrast to other proteins used in formulation science (e.g., albumin), lysozyme, with a high isoelectric point (pI) of 10.7, and a net positively charged protein at physiological pH (7.4), can strongly bind to the negatively charged MSNs surface via electrostatic interaction and further prolong the drug-release rate [29].

We herein report the development of UCNPs and MSN-based hierarchical nanostructure as smart DRSs. As illustrated in Figure 1, the designed DRSs consist of four key components: (i) UCNPs (~four NPs) are the core of the composite nanostructure, responsible for interacting with NIR light (980 nm) to emit UV light; (ii) the three-dimensional (3D) cubic mesostructured silica shell carries drug molecules and molecule stirrer; (iii) the molecule stirrer loaded into the pores of the silica shell for absorbing UV/vis emitted from UCNPs creates a continuous rotation–inversion movement, which act as a molecular impeller that propels the drug release [30]; and (iv) lysozyme is coated onto the silica shell to prolong the drug release. The efficiency of this designed system was evaluated in breast cancer cells demonstrating the controlled release of a chemotherapy drug from the developed DRSs under the irradiation of NIR light. Hereby reported findings indicate a solid potential of UCNPs, MSN, and lysozyme that can be combined towards the developing of controlled DRSs.

## 2. Materials and Methods

### 2.1. Materials and Agents

Unless otherwise specified, both the reagents for cell culture and reagents employed for non-cellular protocols were purchased from Sigma-Aldrich (Truganina, VIC, Australia).

### 2.2. Fabrications and Coatings of Nanostructures

#### 2.2.1. Synthesis of Mesoporous Silica Nanoparticles (AMS-6)

The synthetic method of AMS-6 has been described in our previous work [31]. Briefly, in this synthesis, the surfactant, N-Lauroyl-L-Alanine (1.25 g), was first added to 250 mL Mill-Q water in a PVC bottle, and the solution was kept at 80 °C and stirred at 400 rpm for 12 h. The stirring was increased to 1000 rpm for 10 min to the surfactant solution, and then, the co-structure directing agent, 3-aminopropyl triethoxysilane (1.25 g APES), and the silica source, TEOS (6.25 g), were added into the surfactant solution. The mixed solution was stirred at 1000 rpm for 1 h, then the stirring rate was reduced to 500 rpm and was maintained for 12 h. The reaction solution was kept at 80 °C. The reacted solution then was kept at room temperature for another 12 h. The as-synthesized AMS-6 nanoparticles were filtered, washed, and dried overnight at RT. Further calcination at 550 °C for 3 h in flowing air was carried out to remove the surfactant, and the final mesoporous nanoparticles were obtained.

#### 2.2.2. Synthesis of Upconversion Nanoparticles (UCNPs)

The typical synthesis procedure of NaYF_4_: 20%Yb: 0.5%Er is as follows [32]: Lanthanide chloride (1 mmol), including YCl_3_, YbCl_3_, and ErCl_3_, was dissolved in methanol at a molar ratio of 79.5:20:0.5 and then mixed with 6 mL oleic acid and 15 mL octadecene. In order to remove methanol and dissolve the lanthanide salts, the mixture was heated to 150 °C for 30 min in a round-bottom flask under the protection of high-purity Ar gas. After cooling to RT, the methanol solution containing 2.5 mmol sodium hydroxide (NaOH) and 4 mmol ammonium fluoride (NH_4_F) was added into the reaction flask, and the mixed solution was stirred for another 30 min at RT. Then, the mixture was heated to 90 °C and kept for 30 min and continued to heat to 150 °C; the solution was maintained at this temperature for 10 min to remove the water and methanol. Subsequently, the reaction solution was further heated to 300 °C and kept at this temperature under gentle stirring for 90 min. The nanoparticles were washed using an oleic acid, cyclohexane, methanol, and ethanol mixture after the reaction solution was cooled to RT. The obtained UCNPs were dispersed in cyclohexane.

#### 2.2.3. Synthesis of the Core@shell Nanostructure of UCNPs@AMS-6

In the synthesis procedure, the surfactant, *N*-Lauroyl-l-Alanine (0.14 g), and the UCNPs dispersion (2 mL, 10 mg/mL) were firstly added to 26 mL Mill-Q water in a PVC bottle, and the solution was kept at 80 °C while stirring at 400 rpm for 12 h. The stirring was increased to 1000 rpm for 10 min to produce the surfactant solution, and then, the APES (965 µL) and TEOS (135 µL) were added into the reaction solution in step. Following the synthesis procedure of the AMS-6 nanoparticles, the as-synthesized UCNPs coated with AMS-6 (UCNPs@sAMS-6) were filtered, washed, and dried overnight at RT. The calcination was carried out at 400 °C for 3 h in flowing air to remove surfactant and preserve UCNPs’ luminescent property, and the UCNPs@cAMS-6 nanoparticles were obtained.

#### 2.2.4. Drug and Molecule Stirrer Loading

A solvent evaporation procedure was applied to load doxorubicin hydrochloride (DOX) into the UCNPs@cAMS-6. In brief, the UCNPs@cAMS-6 (30 mg) was added into Mill-Q water (100 mL) in a round-bottom flask while stirring for 20 min with sonication. The DOX (6 mg) and molecule stirrer (MS, 1.5 mg, 7-[(4-Hydroxyphenyl)diazenyl]naphthalene-1,3-disulfonic acid) were added into the dispersion of the UCNPs@cAMS-6, then the dispersion was stirred gently for 1 h. The DOX- and MS-loaded UCNPs@cAMS-6 was obtained using rotary evaporation (100 rpm, 40 °C, and 50 Pa) to remove Mill-Q water. The powder sample was dried at atmospheric pressure for 1 h, and stored in airtight containers. With a varied ratio of the DOX, MS, and core@shell nanoparticles, the loading amount was optimized. A low loading capacity was specifically targeted in order to avoid the drugs crystallizing.

#### 2.2.5. Lysozyme Coating onto the Drug-Loaded UCNPs@cAMS-6 Nanostructures

UCNPs@cAMS-6@DOX&MS nanoparticles (1 mg) were incubated with the lysozyme solution (1 mL, 50% *w*/*w*) for 10 min at 37 °C. The extra lysozyme was removed by centrifuging the reaction solution for 15 min at 13,200 rpm (RT). The supernatant was discarded, and the remaining nanoparticles pellets were re-dispersed in water. This step was repeated three times to remove the loosely absorbed lysozyme. The sample was dried using a freeze dryer (Martin Christ Freeze Dryers, 1200 mL) for lyophilization at −50 °C and 0.1 mbar for 24 h. The lysozyme-coated UCNPs@cAMS-6@DOX&MS was obtained as UCNPs@cAMS-6@DOX&M- S@Lyso.

### 2.3. Characterization, Measurements and In Vitro Cell Assessments

#### 2.3.1. Physical and Chemical Properties Characterizations

Powder X-ray diffraction (Bruker D8 Discover Diffractometer) studies were performed on the drug-free and drug-loaded nanostructures to evaluate the crystallinity of the AMS-6, UCNPs, and DOX. The X-ray source was Cu-Kα (λ = 1.5406 Å), the scanning range was 0–70°, and the scanning step was 0.02°.

Textural properties were characterized using N_2_-adsorption desorption isotherms measurement on all samples at liquid nitrogen temperature (−196 °C) using a Micromeritics TriStar II volumetric adsorption analyzer. Before the measurements, all samples were dried and degassed for 12 h at 40 °C and 100 °C for drug-loaded and drug-free samples, respectively. Specific surface areas of samples were calculated using the Brunauer–Emmett–Teller (BET) method in the relative pressure range of 0.05~0.2. The total pore volume was considered at P/P_0_ = 0.95.

Thermogravimetric analysis was carried out to determine the loading amount of the drug in mesoporous silica carrier using a TGA-2050 (TA instruments, New Castle, DE, USA). The heating temperature ramp was from 20 to 700 °C at a heating rate of 20 °C·min^−1^. The sample weights varied from 5 mg to 10 mg.

SEM (Zeiss Supra 55 at 5 KV) and TEM (FET Tecnai T20 at 200 KV) were adopted to observe morphology and topography of the nanostructures. Samples were coated with a thin layer of gold coating for SEM.

Photoluminescent spectra (PL) were recorded of the distilled water dispersions of the as-synthesized UCNPs@sAMS-6 and calcined UCNPs@cAMS-6 using an Ocean Optics QE65000 spectrometer at 980 nm. The emission intensities of the emissions of the nanostructures were normalized to the emission intensity at 650 nm. All experiments were performed with a power density of 2 MW/cm^2^.

Fourier transform IR (FTIR) spectra of the fabricated samples in powder phase were obtained using a Thermo Scientific Nicolet iS5 FT-IR Spectrometer with iD5 ATR accessory, in transmittance. A wavenumber range from 4000 cm^−1^ to 400 cm^−1^ was scanned for 32 repeats for each sample.

The surface charge of the various nanostructures was measured using a Zetasizer ZS at 25 °C with a He-Ne laser (633 nm, 4 mW output power) as a light source. The various nanostructures were dispersed in Mill-Q water (700 µL, 1 mg/mL).

#### 2.3.2. Drug-Release Profile Monitoring

Both UCNPs@cAMS-6@DOX&MS and UCNPs@cAMS-6@DOX&MS@Lyso (1 mg) were added into 50 mL distilled water in separate beakers with parafilm cover and under 50 rpm continuous stirring at 37 °C. The 700 μL solution was carefully taken at specific time points from the top of the beaker to avoid nanoparticles. The taken solutions were transferred into a cuvette to carry out the UV–visible spectrometer measurement to determine the amount of DOX released. After testing, the solution was transferred back into the beaker. Near-infrared light-triggered drug release was conducted using a 980 nm NIR laser at power density of 2.0 W cm^−2^. The NIR laser was set 30 cm away above the top surface of the solution in the beaker. Then, the solution was also collected for analysis at specific time points under the irradiation of the NIR laser for UV–vis spectrometer measurements.

#### 2.3.3. Drug-Release Effect on Breast Cancer Cells

MCF-7 cells from Australia Cell Bank were cultured with RPMI1640 medium (In Vitro Technologies, Australia) in six-well cell culture plates. The medium was supplemented with 10% fetal bovine serum (FBS) and 100 U mL^−1^ penicillin/streptomycin. Cells were incubated in a standard incubator with 5% CO_2_ at 37 °C. When 70–80% confluence was reached, cells were passaged and not used after more than 3–4 passages. The MCF-7 cells were transferred into a 96-well plate with one control group and three experimental groups set up in four separated plates. Before adding the nanoparticle, the cells were cultured for two days. In each well, the nanostructured samples were added with a final concentration of 30 µg mL^−1^. The nanostructured samples were firstly co-cultured with the cells for 1 h. Then, the experimental groups were irradiated using a 980 nm NIR laser with alternative ON/OFF time of 5 min at power density of 2.0 W cm^−2^. After that, the cells were cultured in the incubator for 48 h. The viability of the cells was measured by the Live/Dead Cell Assay kit (Sigma-Aldrich, Australia) with staining for 1 h at 37 °C. Because the calcein AM has the same emission as the doxorubicin, the dead cells were stained with propidium iodide, and the live cells were stained by calcein blue. The cells were imaged under the confocal microscope (FLUOVIEW FV 1200, Olympus Life Science, Shinjuku, Japan) with a 20× objective. All the images were taken under the same excitation intensity.

## 3. Results and Discussions

In order to construct the designed DRSs with a hierarchical structure, one of the most efficient UCNPs [33,34,35], namely NaYF_4_: 20%Yb, 0.5%Er, was firstly synthesized. The morphology of the synthesized NaYF_4_: 20%Yb, 0.5%Er UCNPs was observed using TEM (Appendix A). The TEM images show that the UCNPs were uniform, with an average diameter of 27.5 nm.

The synthesized UCNPs were incorporated with the 3D cubic mesoporous silica in the core@shell structure (designated as UCNPs@AMS-6). The morphology of the formed UCNPs@AMS-6 nanoparticles is shown in Figure 1a,b, exhibiting homogenously spherical shapes and 140 nm size. There were on average four UCNPs integrated into one single UCNPs@AMS-6 nanoparticle. Comparing the morphology of the AMS-6 silica spheres before (Appendix A), and after the incorporation with UCNPs (Appendix A), the shape and size did not vary much, suggesting the solid morphological stability of the UCNPs@AMS-6 nanoparticles. Detailed characterization of the physical properties of the UCNPs@AMS-6 nanoparticles was carried out. Figure 1c shows the normalized photoluminescent spectra of the UCNPs@AMS-6 nanoparticles before the calcination treatment (UCNPs@sAMS-6) and after (UCNPs@cAMS-6). It is evident that both the UCNPs@sAMS-6 and UCNPs@cAMS-6 show the same emission wavelengths at 380 nm, 420 nm, 525 nm, 545 nm, and 650 nm, while the emission intensities at these wavelengths were reduced by 20%, suggesting that the calcination treatment impacted negligibly to the upconversion emission properties [36,37] The isotherm of the calcinated AMS-6 silica (cAMS-6) and UCNPs@cAMS-6 nanoparticles was also measured and compared (Figure 1d). The surface area and an average pore diameter of the cAMS-6 are 716.79 m^2^/g and 46.0 Å (Figure 1d and Appendix A), respectively. After incorporating cAMS-6 with UCNPs, the outmost mesoporous shell structure of the UCNPs@cAMS-6 has an average pore diameter of 43.0 Å and surface area of 446.54 m^2^/g, and we observed a decrease in the surface area while the pore size was maintained, which could be attributed to the UCNPs taking up around 37% of the space of cAMS-6.

Before coating the silica shell with lysozyme, a chemotherapeutic agent, DOX, was loaded together with the MS into the porous silica shell; the pore size distribution of the loaded sample can be faintly discerned, showing the drug and MS successfully and completely filled the pores (Appendix A). Furthermore, the total amount of loaded DOX and MS were quantified via TGA measurements (Appendix A). The loading amount of the DOX and MS in total is 20 wt%, including 16 wt% of DOX and 4 wt% of MS in UCNPs @cAMS-6@DOX&MS.

To achieve the prolonged release, the UCNPs@cAMS-6@DOX&MS nanoparticles were coated with lysozyme to form a protein layer via the electrostatic interaction between lysozyme and the surface of MSN [38]. Appendix A shows the typical bands of the lysozyme in the FTIR spectra, the amide I peak at 1643 cm^−1^, and amide II at 1528 cm^−1^. These vibrations, albeit slightly shifted in frequency and with lower intensity (amide I: 1583 cm^−1^, II: 1496 cm^−1^), are observed in the spectra of UCNPs@cAMS-6@DOX&MS@Lyso nanoparticles. The 1410 cm^−1^ C–H in-plane bending vibration of DOX is easily discernible as well. Furthermore, we carried out the zeta potential measurements for all the nanostructured samples. The surface charge of UCNPs@sAMS-6 is 16.5 mV and UCNPs@cAMS-6 is −15.9 mV, confirming that after calcination the surfactant of *N*-Lauroyl-l-Alanine was removed, and the pure silica surface with –OH groups shows the negatively charged surface (Appendix A). Once the DOX and MS were loaded, the surface charge of the UCNPs@cAMS-6@DOX&MS decreased to −2.62 mV, suggesting that the DOX and MS molecules can strongly attach to the mesopores of the UCNPs@cAMS-6 through formation of strong hydrogen bonds and charge interactions with the surface silanol groups. When drug-loaded mesoporous silica was incubated with lysozyme, the surface charge changed to 7.04 mV, suggesting the lysozyme successfully conjugated onto the mesoporous surface. The compacted lysozyme was expected to entrap the drug molecules inside the pores and prolong the drug release effectively.

The drug-release profiles in an aqueous environment at pH 7.0 over 7 h from the nanostructured samples with and without lysozyme coating and the irradiation using a 980 nm NIR laser are shown in Figure 2a. The NIR 980 nm laser was employed as an external stimulation to control the drug release. UV–vis absorption spectroscopy was used for monitoring the concentration variation of DOX in the drug-release process and the calibration curve of the DOX concentration against the absorption peak value presents the two parameters in a linear, incremental relationship (Appendix A).

Interestingly, without the lysozyme coating, the drug release from the UCNPs@cAMS-6@DOX&MS with or without NIR light excitation increased quickly in the first 30 min, and the normalized release ratio reached to 0.42 (42%) and 0.25 (25%), respectively (Figure 2a, i and ii). This suggests that there is a burst release without lysozyme coating. The drug release from the UCNPs@cAMS-6@DOX&MS nanoparticles under the NIR excitation achieved 0.92 (92%) within 3 h (Figure 2a, i). Comparatively, The DOX release from the UCNPs@cAMS-6 @DOX&MS nanoparticles without the NIR light irradiation exhibited a lower ratio of drug release of 0.74 (74%) within 3 h (Figure 2a, ii), which demonstrated NIR excitation is an effective way to promote the drug release.

Figure 2a, iii and iv display the drug release from the UCNPs@cAMS-6@DOX &MS@Lyso nanoparticles without the NIR light irradiation in the first 4 h. The burst release of the drug at the beginning stages was markedly prohibited due to the lysozyme coating. The drug-release ratio for the two batches’ measurement was similar at 0.52 (52%), while the ratio of drug release reached to 0.81 (81%) in 4 h for the sample without lysozyme coating (Figure 2a, ii), indicating the drug was entrapped efficiently by the compact block protein.

Moreover, the effect of the NIR light irradiation on the controlled release was further confirmed by applying the irradiation for the next 6 h to the same batches of the UCNPs @cAMS-6@DOX&MS@Lyso nanoparticles. Firstly, the UCNPs@cAMS-6@DOX&MS@Lyso nanoparticles without NIR light show a drug-release profile of around 0.5 (50%) (Figure 2b, i). Once NIR light was applied after 4 h, the drug release of the UCNPs@cAMS-6@DOX&MS@Lyso nanoparticles started to accelerate, and the drug-release amount was higher than the sample without irritation in the 6th hour. At the end of 10 h, with the irradiation of NIR light, 83% of DOX was released (Figure 2b, iii). The release acceleration rate (k) was calculated roughly at 0.09, which is significantly lower than that of nanostructured sample without protein coating under NIR irradiation (k ≈ 0.31) (Figure 2a, i), which further confirms that the protein coating can effectively prolong the drug-release process.

The total released drug amounts over 10 h for the samples with lysozyme coating were still lower than that from the UCNPs@cAMS-6@DOX&MS without lysozyme both with and without NIR light, suggesting that lysozyme coating and NIR lighting can evolve into one single drug model to control DRSs. These results demonstrate that we can develop a wide variety of NIR-responsive DRSs, where the NIR laser and lysozyme can effectively modulate DRSs, both actively promoting and prolonging the drug-release kinetics.

The preliminary in vitro evaluation of the designed DRS effect was carried out using a specific breast cancer cell line, MCF-7. Figure 3 displays the confocal microscopy images of the MCF-7 cells viability after co-cultured with either UCNPs@cAMS-6 @DOX&MS or UCNPs@cAMS-6@DOX&MS@Lyso nanoparticles at 37 °C for 48 h in the presence or absence of NIR light irradiation. Cells stained with green using calcein AM represent live cells, whereas cells stained with red using propidium iodide are representative of dead cells. We firstly assessed the irradiation effect of NIR light (980 nm, 2 W·cm^−2^, and 30 cm from laser to top of the plate) on the cells without the drug or any nanoparticles present (Figure 3a–c), suggesting that the applied NIR laser has no observable phototoxicity to the cells. In the absence of NIR light, both the UCNPs@cAMS-6 and UCNPs@cAMS-6@Lyso exhibited negligible cytotoxicity on MCF-7 cells (Appendix A). Without NIR light, the cytotoxic effect of the UCNPs@cAMS-6@DOX&MS nanoparticles in Figure 3d–f was more pronounced than following treatment with the nanoparticles containing lysozyme coating (Figure 3g–i). The quantitative analysis of the cell viability also showed the same trend (Appendix A). This finding confirmed that the protein coating is indeed capable of prolonging the drug-release kinetics. On the other hand, with the NIR light irradiation, the DRS based on UCNPs@cAMS-6 @DOX&MS nanoparticles exhibited the lowest cell viability (Figure 3j–l) with a fluorescence intensity of 0.94 (Appendix A), which is consistent with the drug-release profiles in Figure 2a and demonstrates that the NIR light is capable of promoting the drug release. In relation to the lysozyme-coated nanoparticles (UCNPs@cAMS-6@DOX&MS@Lyso), the red fluorescence intensity was 0.29 (Appendix A), which is higher than that of the same nanostructured samples without NIR light but lower than that of the nanostructured samples without lysozyme coating. Overall, the most pronounced effect was observed from the UCNPs@cAMS-6@DOX&MS under NIR light (Figure 3j–l), followed by UCNPs@cAMS-6@DOX&MS@Lyso under NIR light (Figure 3m–o). It is evident that the designed DRS containing UCNPs @cAMS-6@DOX&MS@Lyso can effectively be applied for cancer cell treatment via the controlled drug release by using lysozyme and NIR light irradiation, respectively.

## 4. Conclusions

In summary, we designed and fabricated a smart DRS containing UCNPs@ cAMS-6@DOX&MS nanoparticles that can achieve increased or prolonged drug release in one nano-system. The protein lysozyme coated onto the hybrid system successfully reduced the free release of the drug molecules, while reducing the release rate to achieve the prolonged drug-delivery process; Meanwhile, the promoted drug release can be effectively achieved under NIR light irradiation, because the UCNP will effectively absorb the NIR light and emit the UV–vis light and, while the molecular stirrer within the drug model, can adsorb the UV–vis and create a continuous rotation–inversion movement, which leads to an increase in the drug release. Moreover, the smart DRS showed negligible cytotoxicity and demonstrated effective capability to release the chemotherapy agent for cancer cell treatments. The design of an efficient DRS presented here demonstrates a proof-of-concept that will shed new light onto the future design and applications of multifunctional platforms for cancer theranostics.

## Data Availability

Data supporting the findings of this study are available from the corresponding author upon reasonable request.

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
