# Peer review of "Smart Drug-Delivery System of Upconversion Nanoparticles Coated with Mesoporous Silica for Controlled Release"

_pharmaceutics, 2022, doi:10.3390/pharmaceutics15010089_

Round 1

Reviewer 1 Report

The present manuscript entitled " Smart drug delivery system of upconversion nanoparticles coated with mesoporous silica for controlled release " described a creation of a drug release system controlled  by external stimulation by near-infrared light (NIR) and coated with a protein.

The study is quite interesting due to the implication of the UCNPs which have an enormous potential for biological applications. In addition, Yanan Huang et al. make an in-vivo therapeutic study on model cancer cells that brings more dimensionality to the manuscript. In general, the work reads fluently and clear, but there are some points that need to be clarified before its publication.

Firstly, I would recommend a thorough proof-reading of the article for minor grammar errors or typos, for instance:

Page 2 line 26 for the loading OF drug molecules,

Page 4 line 142 drug and Molecule stirrer loading

Page 6 line 233 there WERE on average 4 UCNPs integrated into…

Etc.

Secondly, in Figure 1 c it is obvious that the spectrum after the process of calcination has changed to a different extent for the bands at 400, 550 and 650 nm. How do authors explain this? There is plenty of literature explaining the different possible influences on the band intensity ratios (e.g. http://dx.doi.org/10.1039/C5RA11502G)

Also. The aggregation obvious from the Figure 1 a and b was not addressed – was it an impediment for the bio applications? How big were the overall clusters? How can the clustering be reduced or overcome?

Next, on page 8, lines 283 on the authors describe the process of the drug release experiment “The drug release profiles in an aqueous environment at pH 7.0 for 7 hours from the 283 nanostructured samples with and without lysozyme coating and the irradiation of a 980 284 nm NIR laser”. Was the laser pulsed or continuous-wave? Did the authors address the issue of overheating, given the high energies and long exposure used in the experiment? The NIR lasers are known for overheating the samples overtime, so this might be an issue for the credibility of the drug release experiment, since it can also be a factor to be considered. I would suggest the authors to include another control sample to the graph in Figure 2 (A), where one or two of the samples (say, with and without the protein coating) are incubated in water without irradiation.

Author Response

The present manuscript entitled " Smart drug delivery system of upconversion nanoparticles coated with mesoporous silica for controlled release " described a creation of a drug release system controlled by external stimulation by near-infrared light (NIR) and coated with a protein.
The study is quite interesting due to the implication of the UCNPs which have an enormous potential for biological applications. In addition, Yanan Huang et al. make an in-vivo therapeutic study on model cancer cells that brings more dimensionality to the manuscript. In general, the work reads fluently and clear, but there are some points that need to be clarified before its publication.

We are grateful for the time and effort the reviewer has spent in reviewing our manuscript. The review comments are very helpful for further strengthening the manuscript.

1.Firstly, I would recommend a thorough proof-reading of the article for minor grammar errors or typos, for instance:

Page 2 line 26 for the loading OF drug molecules,

Page 4 line 142 drug and Molecule stirrer loading

Page 6 line 233 there WERE on average 4 UCNPs integrated into…Etc.

Reply: Thanks for the reviewer’s careful review. The typos and syntax errors have been corrected through the whole manuscript in red font and the corrections are listed as below:

Page 2 line 26 for the loading of drug molecules;

Page 4 line 142 Drug and molecule stirrer loading

Page 4 line 143 A solvent evaporation procedure was applied

Page 5 line 191 The surface charge of the various nanostructures was measured

Page 6 line 225 To construct the designed DRSs with a hierarchical structure,

Page 6 line 232 There were on average 4 UCNPs integrated

2.Secondly, in Figure 1 c it is obvious that the spectrum after the process of calcination has changed to a different extent for the bands at 400, 550 and 650 nm. How do authors explain this? There is plenty of literature explaining the different possible influences on the band intensity ratios (e.g. http://dx.doi.org/10.1039/C5RA11502G)

Reply: Thanks for the reviewer’s question. The reviewer is right about the changes of the emission intensities of the nanostructure to slightly different extents. The mesoporous structure of the silica was achieved via a calcination process at 400 °C for 3 hrs to remove surfactant after UCNPs were incorporated as the core. During the calcination, the crystal structure of UCNPs would be inevitably damaged from the high temperature (Padmaha Parameswaran et al. Biomaterials Advances 2022, 136, 212763). The crystal structure of the UCNPs strongly affects the photon energy transfer from Yb3+ to Er3+, which result in the decreased emission intensities. The emission at the wavelength 420 nm is a four-photon emission process, while the emissions at 575 nm and 650 nm are three-photon and two-photon emission process respectively. The slightly damage of the crystal integrity of the UCNPs would affect these photo energy transfer processes to different extents, leading to emission intensity variation at different extents.

We did optimization on the calcination temperatures for maximized pore volume and well-maintained upconversion luminescent property of the UCNPs. Figure R1 show the emission spectra (a & c) and TEM morphologies (b & d) of the UCNPs@cAMS-6 calcinated for 3 hrs at 450 °C and 500 °C, respectively. The results show that as the calcination temperature increased, the emission intensities at different wavelengths decreased much more dramatically, suggesting 400 °C was the optimal calcination temperature both for remove the surfactant from as-synthesized silica as well as maintain the properties of the UCNPs.

Thanks to the reviewer’s suggestion, we have revised the corresponding descriptions in the manuscript and added the reference.

(Image R1 can be find from the attached PDF doc)

Figure R1 The luminescent emission spectra (a & c) and TEM morphologies (b & d) of the UCNPs@cAMS-6 calcinated for 3 hrs at 450 °C and 500 °C, respectively.

3.Also. The aggregation obvious from the Figure 1a and b was not addressed – was it an impediment for the bio applications? How big were the overall clusters? How can the clustering be reduced or overcome?

Reply: Thanks for reviewer’s question. We agree with the reviewer that the aggregation of the UCNPs@cAMS-6 nanoparticles and the agglomeration would be an impediment factor for further biomedical applications. The samples for SEM and TEM characterizations were prepared from dry power samples which were stocked from the synthesis. Hence, it is easy to find the agglomerated mesoporous silica because of the Si-O-Si bond on their surfaces.

The average size of the overall clusters is roughly 600 nm from TEM characterization. In SEM images, we can see much larger aggregates, it is because we pour the power samples onto the carbon tape directly without dispersing the nanoparticles in ethanol. 

However, much better dispersed mesoporous silica nanoparticles can be achieved by dispersing them in ethanol and sonicate for more than 15 mins. In this work, we did disperse the silica nanoparticles in the Mill Q water and sonication for 30 minutes to maximize the homogenous dispersibility of mesoporous silica nanoparticles. We will, in our next work, investigate the size and aggregation of the mesoporous silica coated nanoparticles and to evaluate their effects on the drug delivery and release, and further to evaluate the effects on biomedical applications.

4.Next, on page 8, lines 283 on the authors describe the process of the drug release experiment “The drug release profiles in an aqueous environment at pH 7.0 for 7 hours from the nanostructured samples with and without lysozyme coating and the irradiation of a 980 nm NIR laser”. Was the laser pulsed or continuous-wave? Did the authors address the issue of overheating, given the high energies and long exposure used in the experiment? The NIR lasers are known for overheating the samples overtime, so this might be an issue for the credibility of the drug release experiment, since it can also be a factor to be considered. I would suggest the authors to include another control sample to the graph in Figure 2 (A), where one or two of the samples (say, with and without the protein coating) are incubated in water without irradiation.

Reply: Thanks for reviewer’s careful check and the constructive suggestion. It is a concern that the overheating can be caused from long time exposure to NIR irradiation. With the same concern on overheating from irradiation, we did the pre-check of the NIR light irradiation on pure Milli Q water with and without the UCNPs@cAMS-6@DOX&MS@Lyso nanoparticles. There was no temperature variation after the long-term irradiation for both samples. It was mainly because the NIR light source was 30 cm away above the top level of the liquid, and the irradiation power density we used was 2.0 W cm-2.

Secondly, there were control samples in drug release profile monitoring results in Figure 2 to verify the effect of the overheating. In Figure 2(a), there were four scenarios,

1, UCNPs@cAMS-6@DOX&MS without lysozyme and without irradiation (red line (ii)),

2, UCNPs@cAMS-6@DOX&MS without lysozyme and with irradiation (black line (i)),

3, UCNPs@cAMS-6@DOX&MS with lysozyme and without irradiation (pink line (iv)),

4, UCNPs@cAMS-6@DOX&MS with lysozyme and without irradiation in first 4 hours and with irradiation for further 6 hours (blue line (iii)).

Comparing 1 to 2 is to evaluate the NIR light irradiation effect on the drug release from the nanostructures without lysozyme coating, and the comparison result shows the NIR irradiation increases the drug release. Comparing 1 and 3 is to investigate the effect of lysozyme effect on the drug release from the nanostructures, and the comparison result demonstrates the lysozyme coating indeed slow down the drug release.

The comparison between 3 and 4 exhibited the slowing down effect of lysozyme in the first 4 hours with no irradiation, and also showed the increased drug release under NIR light irradiation for the nanostructures with lysozyme coating. The release rate difference between the irradiation ON and OFF can further proof that the NIR light irradiation can effectively promote the drug release even with the lysozyme coating. All samples in dispersion under NIR light irradiation were found no overheating and the effect from the heat on drug release can be omitted.

Reviewer 2 Report

The manuscript deals with the interesting fields of smart systems for drug release and is focused on the possibility of introducing two different tools for controlling drug release. It is interesting and deserves to be published with  minor modifications.

Questions/suggestions

Pag 5, lines 143-145: this part should be better explained: the amount of water should be indicated, moreover when DOX chloridrate is added, was it  in solution or a suspension?

Paragraph 2.2.4.: this kind of procedure is called solvent evaporation (not wet incipient). I suggest to revise.

What do the Authors mean for  drug efficiency loading? Which formula do they use to determine the loading efficiency? If they mean the amount of loaded drug, it is called drug loading and is expressed as amount of drug present in the prepared system (% w/w). Efficiency means the amount of the loaded drug in comparison to the total added drug during the loading procedure.

The same problem in lines 261-262.

Line 159: what do you mean, absorb or adsorb? why do the Authors want to eliminate the adsorbed lisozyme? I presume the not adsorbed lisozyme. Please make it clear.

Lines 191: does it mean that the spectra were recorded in solid state?

Line 199: beak or beaker? Please check.

Line 283: deionized water was used, as indicated in experimental part. Did you measured the pH? The pH of deionized water is usually slightly acid.

Figure 2. y.axis is not a percentage, but a normalized DOX release ratio.  In the text, percentage of release is described. Why different names are used in the text? You should uniform it.

Line 314-316. How was calculated the release acceleration rate (k)?

Lines 333: “To verify the desired therapeutic effect from the controlled DRS, we adopted a specific breast cancer cell line (MCF-7) as an example”: The study is in vitro, thus the therapeutic effect cannot be evaluated in this way. It is a preliminary evaluation of in vitro activity. Please, revised this sentence.

Too many figures have been reported in supplemental information. Thus the Authors should choose at last two or three meaningful figures and add them in the manuscript.

Author Response

The manuscript deals with the interesting fields of smart systems for drug release and is focused on the possibility of introducing two different tools for controlling drug release. It is interesting and deserves to be published with minor modifications.

We are grateful for the time and effort the reviewer has spent in reviewing our manuscript. The review comments are very helpful for further strengthening the manuscript.

1.Page 5, lines 143-145: this part should be better explained: the amount of water should be indicated, moreover when DOX chloridrate is added, was it in solution or a suspension?

Reply: Thanks for reviewer’s careful revision. The description of the line 143-145 in Page 5 has been revised.

Page 5 line 143 – 145:

“A solvent evaporation procedure was applied to load doxorubicin hydrochloride (DOX) into the UCNPs@cAMS-6. In brief, the UCNPs@cAMS-6 powder (30 mg) was added into Mill-Q water (100 mL) in a round-bottom flask and stirred for 20 min followed with sonication for 30 mins. The DOX powder (6 mg) and molecule stirrer powder (MS, 1.5 mg, 7-[(4-Hydroxyphenyl)diazenyl]naphthalene-1,3-disulfonic acid) were added into the dispersion of the UCNPs@cAMS-6 in the round-bottom flask, then the dispersion was stirred gently for 1 hour.”

2.Paragraph 2.2.4.: this kind of procedure is called solvent evaporation (not wet incipient). I suggest to revise.

Reply: Thanks for reviewer’s kind suggestion. We have corrected the procedure as follow: 

Page 5 line 143:

“A solvent evaporation procedure was applied to load doxorubicin hydrochloride (DOX) into the UCNPs@cAMS-6.”

3.What do the Authors mean for drug efficiency loading? Which formula do they use to determine the loading efficiency? If they mean the amount of loaded drug, it is called drug loading and is expressed as amount of drug present in the prepared system (% w/w). Efficiency means the amount of the loaded drug in comparison to the total added drug during the loading procedure. The same problem in lines 261-262.

Reply: Thanks for reviewer’s question and kind suggestion. We carefully corrected the expressions in the manuscript.

Page 2 line 46: (i) a biocompatible nanocarrier with high drug loading amount,

Page 4 line 151: the loading amount was optimized.

Page 6 line 260: The loading amount of the DOX and MS in total is 20 wt%;

4.Line 159: what do you mean, absorb or adsorb? why do the Authors want to eliminate the adsorbed lisozyme? I presume the not adsorbed lisozyme. Please make it clear.

Reply: Thanks for reviewer’s careful revision. We apologize for the confusing words of adsorb and absorb. The reviewer is right and it should be absorbed lysozyme. We want to remove the loosely absorbed lysozyme on top of lysozyme to ensure the lysozyme was coated directly on the surface of the nanoparticles, hence, the washing step was repeated for three times to remove the loosely coated lysozyme. The description is corrected as below:

Page 4, line 158 - 159:

“This step was repeated 3 times to remove the loosely absorbed lysozyme.”

5.Lines 191: does it mean that the spectra were recorded in solid state?

Reply: Thanks for reviewer’s question. The Zeta potential measurement for surface charge of the various nanostructures were conducted for a variety of nanostructures dispersed in Milli Q water in the concentration of 1mg/mL. Therefore, it was not recorded in solid state. The description of the zetapotential measurement has been revised as below:

Page 5 line 191 - 194: 

“The surface charge of the various nanostructures was measured using a Zetasizer ZS at 25 ֯C with a He-Ne laser (633 nm, 4mW output power) as a light source. The various nanostructures were dispersed in Mill-Q water (700 µl, 1mg/mL).

6.Line 199: beak or beaker? Please check.

Reply: Thanks for pointing this out. It should be “beaker”. We have revised in the manuscript in red font.

7.Line 283: deionized water was used, as indicated in experimental part. Did you measured the pH? The pH of deionized water is usually slightly acid.

Reply: Thanks for reviewer’s careful review. We actually used Mill-Q water in all our experiments and the pH was measured as 6.98. We apologize for the unprecise word selection.  We have corrected all the “deionized water” to “Mill-Q water” in the manuscript.

Page 3 line 101: “Mill-Q water in a PVC bottle and the solution”

Page 4 line 134: “were firstly added to 26 ml Mill-Q water in a PVC”

Page 4 line 145: “Mill-Q water (100 mL) in a round bottom flask”

Page 4 line 149: “to remove Mill-Q water.”

8.Figure 2. y.axis is not a percentage, but a normalized DOX release ratio.  In the text, percentage of release is described. Why different names are used in the text? You should uniform it.

Reply: Thanks for reviewer’s careful review and kind suggestion. We have corrected the relevant descriptions in the manuscript to ensure the values consistent. The revised values are in red font in manuscript.

Page 7 line 291:

“quickly in the first 30 min and the release ratio reached up to 0.42 and 0.25

Page 7 line 294:

“achieve 0.92 within 3 hrs (Figure 2a, i).”

Page 7 line 296:

“lower ratio of drug release of 0.74 within 3 hours”

Page 7 line 301 - 302:

“The drug release ratio for both the two batches measurement was similar at 0.52 while the ratio of drug release reached to 0.81 in 4 hours for the sample”

Page 7 line 309:

“drug release profile of around 0.5 (Figure 2b, i).”

9.Line 314-316. How was calculated the release acceleration rate (k)?

Reply: Thanks for reviewer’s question. The “release acceleration rate (k)” was calculated using the slope of drug release ratio against time. For specific time point, based on the data in Figure 2, the slope can be obtained by dividing the difference of the drug release ratios at two time points (values in Y axis) using the time difference (hours in X axis).

10.Lines 333: “To verify the desired therapeutic effect from the controlled DRS, we adopted a specific breast cancer cell line (MCF-7) as an example”: The study is in vitro, thus the therapeutic effect cannot be evaluated in this way. It is a preliminary evaluation of in vitro activity. Please, revised this sentence.

Reply: Thanks for reviewer’s suggestion. We corrected the description for the in vitro work as below:

Page 8 line 332 - 333: “The preliminary evaluation of in vitro activity for the designed DRS was carried out using a specific breast cancer cell line (MCF-7) as an example.”

11.Too many figures have been reported in supplemental information. Thus the Authors should choose at last two or three meaningful figures and add them in the manuscript.

Reply: Thanks for reviewer’s kind suggestion. Our manuscript submitted as the “communication”, based on the requirement of the Pharmaceutics, we selected the most important 4 combined Figures in the manuscript with all supporting data included in SI.

Reviewer 3 Report

1. A good spell check on the whole manuscript is warranted.

2. From the release profile graphs presented in figure 2(a)(i, ii), it looks like in the absence of lysozyme, 60% of the payload is released without NIR stimulation compared to 80% in the presence of stimulation. For a proper stimuli-responsive release these difference needs to be much higher in order to get any real application.

3. The reason for using lysozyme coating is not clear. Looking into the release profile of the model molecule DOX, Figure 2(a) (iii, iv) it looks like lysozyme is just reducing the total amount of payload release and it does not have any effect on the response to stimulation or in the rate of release.

4. In the view of results presented in this manuscript, the conclusion needs to be modified. Basically, the particles without stimulation and without lysozyme coating provide the best control release profile for doxorubicin compared to others.

Author Response

Comments and Suggestions for Authors

1.A good spell check on the whole manuscript is warranted.

Reply: Thanks for the reviewer’s kind suggestion! We have carefully read through the manuscript and corrected all the typos and language mistakes.

2.From the release profile graphs presented in figure 2(a) (i, ii), it looks like in the absence of lysozyme, 60% of the payload is released without NIR stimulation compared to 80% in the presence of stimulation. For a proper stimuli-responsive release these difference needs to be much higher in order to get any real application.

Reply: Thanks for reviewer’s careful review and kind suggestion. The sample of UCNPs@cAMS-6@DOX&MS without stimulation exhibit 60% release, while the same sample with stimulation shows the 80% drug release within the first hour, which enhanced 33.3% of release efficiency compare with the drug model not expose to the NIR light. We agree with the reviewer that the release difference between these two models is not significantly high, which might not be high enough to apply in clinic applications. However, this difference has demonstrated the effect of the NIR light irradiation on the drug release control and this work is the proof of concept for stimulated drug release using NIR light. We are now still pursuing improved control on drug release using better designed nanostructures. For example, we are working on a core@shell@shell structure of UCNPs@dense Silica@mesoporous silica to protect UCNPs from water quenching in aqueous solution. We also try to entrap more UCNPs within one silica nanoparticles to increase NIR light response to enhance the stimulation for drug release. 

We thank reviewer’s comments and will keep working on the structure and materials to achieve the improved outcome in near future.

3.The reason for using lysozyme coating is not clear. Looking into the release profile of the model molecule DOX, Figure 2(a) (iii, iv) it looks like lysozyme is just reducing the total amount of payload release and it does not have any effect on the response to stimulation or in the rate of release.

Reply: Thanks for the reviewer’s question. The drug model in this work is aim to design a smart drug delivery system that can both achieve promoted and prolonged drug release. The lysozyme coated onto silica shell as the most exterior layer is designed to inhibit the burst release in the first hour and prolong the drug release in a more sustainable way.

From the Figure 2a, it can be seen that the lysozyme coated nanostructures released much lower drugs in the first one hour compared to the nanostructures without lysozyme coating.  Figure 2b exhibited obviously increased drug release under NIR light irradiation even with lysozyme coating, suggesting that the lysozyme coating did not affect the NIR light irradiation simulated drug release. Furthermore, the lysozyme coating would also improve the biocompatibility of the nanostructures.

Therefore, we believe the lysozyme coating can play important role in this designed drug release system.

4.In the view of results presented in this manuscript, the conclusion needs to be modified. Basically, the particles without stimulation and without lysozyme coating provide the best control release profile for doxorubicin compared to others.

Reply: Thanks for the reviewer’s helpful comment. Accordingly, we have modified the conclusion to better highlight the key point of this work.

Round 2

Reviewer 1 Report

The authors answered all my questions, and I recommend to publish the paper.